



# ESD Ideas: A Global Warming Scaling Law

*Mikhail Y. Verbitsky[1,2] and Michael E. Mann[3]*

[1]Gen5 Group, LLC, Newton, MA, USA
[2]UCLouvain, Earth and Life Institute, Louvain-la-Neuve, Belgium
[3]The Pennsylvania State University, Department of Meteorology and Atmospheric Science, University Park, PA, USA
Correspondence: Mikhaïl Verbitsky (verbitskys@gmail.com)

**Abstract.** In this study, we highlight a component of global warming variability, a scaling law that is based purely on fundamental physical properties of the climate system. We suggest that three similarity parameters define
the system response to external forcing, and an argument of physical similarity with observed climate responses in the past can be made when all three parameters are identical for the current and historical climates. We determined that the scaling law of global warming is the $(\lambda + 1 + m)$ - power of time, where $\lambda$ is prescribed by external forcing and $m$ is defined by climate system internal dynamics. When the climate system develops in the direction of intensified positive feedbacks, the power $m$ changes from $m = -1$ (negative feedbacks dominate) to $m \geq 1$
(positive feedbacks dominate). We also establish that a "hothouse" climate with dominant positive feedbacks will be preceded by a climate having a property of incomplete similarity in feedbacks similarity parameters. It implies that the same future scenario may be produced by climate feedbacks of different magnitudes as long as their positive-to-negative ratio is the same.

## 1.  Introduction


Dimensional analysis can provide key insights into physical phenomena. It is especially useful when computer resources and observational data are limited, celebrated works of G. I. Taylor (1950) and G. S. Golitsyn (1970) being just two remarkable examples of many. Indeed, the past two decades have seen a considerable expansion in the availability of proxy and instrumental climate observations and the development of increasingly comprehensive
climate models. Yet there are still substantial uncertainties in the upper bound on projected future warming associated with warm climate feedback mechanisms (Mann, 2021). Our motivation for this study, therefore, is to highlight a scaling law that is relevant to future projected warming and is based on fundamental, universal physical properties of the climate system.

We seek a description of the climate system, formulated as the main hypothesis that is comprehensive enough to
characterize a wide variety of climate change scenarios yet includes as few governing parameters as possible. Applying the $\pi$-theorem (Buckingham, 1914) and similarity concepts (e.g., Barenblatt, 2003) to the main hypothesis leads us to a scaling law which can be refined as additional observational data and climate simulation results become available.

## 2.  Methods

**2.1  The main hypothesis and its implication.** Our main hypothesis is as general as the following: The climate system response to anthropogenic radiative forcing depends on both internal dynamics and external forcing. Further, the internal dynamics of the climate system is defined by magnitudes of its positive and negative feedbacks. Thus
our main hypothesis postulates that the global temperature response is a function of time, of feedback timescales, and of the external forcing intensity and shape:

$$T = \varphi(\tau_p, \tau_n, \varepsilon, t, \lambda) \tag{1}$$

Here $T$ is temperature (°C), $t$ is time (sec), $\tau_p$ (sec) and $\tau_n$ (sec) are the timescales of positive and negative feedbacks correspondingly. The external forcing is described as $\varepsilon t^\lambda$, where $\varepsilon$ (°C sec$^{-\lambda-1}$) is the intensity of the external forcing and adimensional parameter $\lambda$ describes external forcing variability; for example, $\lambda = 1$ corresponds to the linear shape of the anthropogenic radiatively-forced global warming trend over the past half century (Mann et al, 2014).

We will now convert our hypothesis (1) into adimensional form using $\pi$-theorem. This transformation is based
on a simple notion that a physical phenomenon does not depend on a choice of the system of units and therefore parameters with independent dimensions may exist only as parts of dimensionless groups. Formally it means that, if we take $\varepsilon$ and $t$ as parameters with independent dimensions, the global temperature response is a function of three similarity parameters, i.e., $\frac{t}{\tau_p}, \frac{t}{\tau_n}$, and $\lambda$:



$$T = \varepsilon t^{\lambda+1} \Phi(\tfrac{t}{\tau_p}, \tfrac{t}{\tau_n}, \lambda) \tag{2}$$

**2.2 A climate with dominant negative feedbacks (complete similarity in $\tfrac{t}{\tau_p}$ ).** When negative feedbacks dominate over positive feedbacks, e.g., $\tfrac{t}{\tau_n} \gg \tfrac{t}{\tau_p}$, and $\tfrac{t}{\tau_p}$ can be neglected (*complete similarity* in the parameter $\tfrac{t}{\tau_p}$, as it has been defined by Barenblatt, 2003), $\Phi(\tfrac{t}{\tau_p}, \tfrac{t}{\tau_n}, \lambda) = \Phi(\tfrac{t}{\tau_n}, \lambda)$ and the scaling law takes the following form:

$$T = k\varepsilon t^{\lambda+1} \left(\tfrac{t}{\tau_n}\right)^m \tag{3}$$

For a given $\lambda$, $k$ and $m$ need to be determined experimentally.

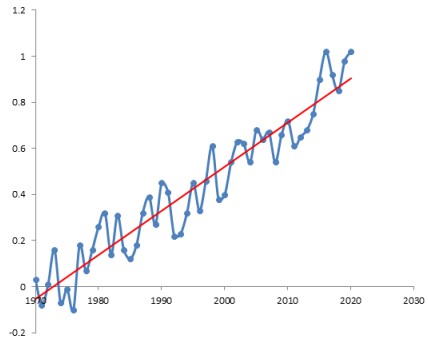


**Figure 1.** Global temperature deviations (°C, GISTEMP Team, 2021) with a linear trendline (red).

The historical global mean surface temperature record can be closely reproduced by a zero-dimensional energy balance model with strong thermal damping (Mann et al, 2014), reflecting the dominance of the (negative) Planck feedback for the range of temperature change observed during the past century. Instrumental measurements (Fig.1)
suggest a linear shape of the historical (1970 – 2020) temperature $T$ evolution ($T \sim t$), under roughly linear anthropogenic (greenhouse gas and sulphate aerosol) radiative forcing ($\lambda = 1$). That implies that $m = -1$ and we arrive to the following scaling law:

$$T = k\varepsilon\tau_n t^\lambda \tag{4}$$

or

$$T \sim t^\lambda \tag{5}$$

**2.3 A climate with dominant positive feedbacks (complete similarity in $\tfrac{t}{\tau_n}$ ).** When positive feedbacks dominate (e.g., under potential "hothouse" scenarios such as explored by Steffen et al, 2018), e.g., $\tfrac{t}{\tau_p} \gg \tfrac{t}{\tau_n}$ , and $\tfrac{t}{\tau_n}$ can be neglected (*complete similarity* in the parameter $\tfrac{t}{\tau_n}$), $\Phi(\tfrac{t}{\tau_p}, \tfrac{t}{\tau_n}, \lambda) = \Phi(\tfrac{t}{\tau_p}, \lambda)$ and:

$$T = k\varepsilon t^{\lambda+1} \left(\tfrac{t}{\tau_p}\right)^m \tag{6}$$

$$T \sim t^{\lambda+1+m} \tag{7}$$

For a given $\lambda$, $k$ and $m$ need to be determined experimentally.



Since empirical data of a "hothouse" climate under well calibrated forcing are not available to us, we will estimate $m$ using analytical solution of the following energy-balance equation for, e.g., $\lambda = 1, \frac{T}{\tau_n} = 0$ and $T(t = 0) = 0$:

$$\frac{dT}{dt} = \frac{T}{\tau_p} - \frac{T}{\tau_n} + \varepsilon t^\lambda \qquad (8)$$


$$T = \varepsilon t^2 \left( \frac{\tau_p^2}{t^2} e^{\frac{t}{\tau_p}} - \frac{\tau_p}{t} - \frac{\tau_p^2}{t^2} \right) \qquad (9)$$

For timescales $\frac{t}{\tau_p} > 1$, $\left( \frac{\tau_p^2}{t^2} e^{\frac{t}{\tau_p}} - \frac{\tau_p}{t} - \frac{\tau_p^2}{t^2} \right)$ can be approximated as $\left( \frac{t}{\tau_p} \right)^m$ where $m \geq 1$.

**2.4 A climate having the property of incomplete similarity in $\frac{t}{\tau_p}, \frac{t}{\tau_n}$.** If the climate system evolves from dominant negative feedbacks ($T \sim t^\lambda$) to dominant positive feedbacks ($T \sim t^{\lambda+1+m}, m \geq 1$), there will be a stage when $m = 0$ and:


$$T \sim t^{\lambda+1} \qquad (10)$$

That means that $\Phi(\frac{t}{\tau_p}, \frac{t}{\tau_n}, \lambda)$ does not depend on $t$. It is possible only if the climate system exhibits either the property of complete similarity in similarity parameters $\frac{t}{\tau_p}, \frac{t}{\tau_n}$, e.g., $\frac{t}{\tau_p}, \frac{t}{\tau_n}$ are negligible and climate does not depend on magnitudes of positive and negative feedbacks (though this is unlikely, we allow for this possibility for the purpose of generality), or it has the property of *incomplete similarity* (Barenblatt, 2003), meaning that none of the parameters can be neglected, but two similarity parameters $\frac{t}{\tau_p}, \frac{t}{\tau_n}$ collapse into a single conglomerate similarity


parameter $V = \frac{1/\tau_p}{1/\tau_n}$ which is the ratio of magnitudes of positive and negative feedbacks.

### 3. Conclusions

We suggest that three similarity parameters define system response to external forcing, i.e., $\frac{t}{\tau_p}, \frac{t}{\tau_n}$, and $\lambda$. An

argument of physical similarity with observed climate responses in the past can be made when all three parameters are identical for the current and historical climates.

We established that the scaling law of global warming is a power law $T \sim t^{\lambda+1+m}$ where $\lambda$ is prescribed by external forcing and m is defined by climate system internal dynamics.

When the climate system develops in the direction of intensified positive feedbacks, the power $m$ changes from

$m = -1$ (negative feedbacks dominate) to $m \geq 1$ (positive feedbacks dominate).

If we associate a "hothouse" climate with dominant positive feedbacks, it will be preceded by a climate of $m = 0$, i.e., the climate of incomplete similarity in parameters $\frac{t}{\tau_p}, \frac{t}{\tau_n}$. This discovery is not intuitive and has significant implications. Most importantly, it implies that for such pre-"hothouse" climate the same future scenario may be produced by climate feedbacks of different magnitude as long as their positive-to-negative ratio $V$ is the

same. The diagnostics of the $V$-number thus is critical for climate prediction, $V \sim 1$ being a signature of the pre-"hothouse" climate.

The response of such pre-"hothouse" climate to the extended linear forcing ($\lambda = 1$) will be time-squared, i.e., $T \sim t^2$.

### Author contributions

MYV conceived the research and developed the formalism. MYV and MEM contributed equally to writing the paper.

### Competing interests

The authors declare that they have no conflict of interest.



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
