# Peer review of "ESD Ideas: A Global Warming Scaling Law"

_Earth System Dynamics, 2021_

## Author Comment (AC5)

Dear Referee 1,

Following your recommendations, we suggest to precede our main hypothesis with the following paragraph (new text is marked red). We hope it will satisfactorily explain our postulate and at the same time connect our reasoning with the existing research.

Mikhail Verbitsky and Michael E. Mann

**2.1 The main hypothesis and its implication.** In the series of papers (Verbitsky et al, 2018, Verbitsky and Crucifix, 2020, 2021, and Verbitsky, 2021) we investigated dynamics of the ice-climate system on the orbital timescales. The model that has been analyzed consists of scaled conservation equations of the non-Newtonian ice flow combined with an energy-balance equation of the global temperature:

$$\frac{dS}{dt} = \frac{4}{5}\zeta^{-1}S^{3/4}(a - \varepsilon F_S - \kappa\omega - c\theta) \tag{1}$$

$$\frac{d\theta}{dt} = \zeta^{-1}S^{-1/4}(a - \varepsilon F_S - \kappa\omega)\{\alpha\omega + \beta[S - S_0] - \theta\} \tag{2}$$

$$\frac{d\omega}{dt} = -\gamma[S - S_0] - \frac{\omega}{\tau} \tag{3}$$

Here $S$ (m$^2$) is the glaciation area, $\theta$ ($^o$C) is the basal ice sheet temperature, and $\omega$ ($^o$C) is the global climate temperature. The profile factor $\zeta$ (m$^{1/2}$) is assumed to be a constant; $a$ (m/s) is snow precipitation rate; $F_S$ is an adimensional, normalized external forcing; $\varepsilon$ (m/s) is the external forcing amplitude; $\kappa$ (m s$^{-1}$ $^o$C$^{-1}$) and $c$ (m s$^{-1}$ $^o$C$^{-1}$) are sensitivity coefficients describing ice mass balance response to $\omega$ and $\theta$; the adimensional coefficient $\alpha$ defines basal temperature response to $\omega$ changes, $\beta$ ($^o$C /m$^2$) and $\gamma$ ($^o$C m$^{-2}$ s$^{-1}$) define the sensitivity of basal temperature $\theta$ and global temperature $\omega$, respectively to the changes of the ice sheet area $S$, $S_0$ (m$^2$) is a reference glaciation area, and $\tau$ (s) is relaxation timescale for $\omega$.

In the system (1) – (3), the ice sheet area makes a positive feedback for the global temperature and the ice sheet basal temperature is a delayed negative feedback for ice dynamics. The most remarkable property of this dynamical system is that on astronomical time scales its dynamics is largely defined by the *V*-number representing a ratio of positive-to-negative feedback magnitudes.

The behavior of the above system is fully described by eight adimensional similarity parameters

$$\pi_1 = \frac{\varepsilon}{a}, \pi_2 = \alpha, \pi_3 = \kappa\gamma\varepsilon T^3, \pi_4 = c\gamma\varepsilon T^3, \pi_5 = \frac{T}{\tau}, \pi_6 = \frac{\gamma T}{\beta}, \pi_7 = \frac{S_0}{\varepsilon^2 T^2}, \pi_8 = \frac{\varsigma}{\varepsilon^{1/2}T^{1/2}}$$

where *T* is the period of the external forcing.

If we replace astronomical timescale *T* with a faster timescale (let say 1-10 years) and adjust other model parameters in such a way that numerical values of similarity parameters remain the same, the dynamics of this fast system will be identical to the behavior of the system (1) – (3) on the orbital time scales.

This reasoning brings us to the intriguing observation. Specifically, if the contemporary climate can be described by the energy-balance equation, like equation (3), and its positive feedback mechanism (that is not, indeed, the land ice any longer) is controlled by a delayed negative feedback (like, for example, it may be the case for the methane (e.g., Dean et al, 2018)), then climate's behavior may be largely governed by magnitudes of its positive and negative feedbacks.

Accordingly, our main hypothesis is as general as the following: The climate system response to anthropogenic radiative forcing depends on both internal dynamics and external forcing. Further, the internal dynamics of the climate system is defined by magnitudes of its positive and negative feedbacks. Thus our main hypothesis postulates that the global temperature response is a function of time, of feedback timescales, and of the external forcing intensity and shape:

$$T = \varphi(\tau_p, \tau_n, \varepsilon, t, \lambda) \tag{4}$$

**Additional references**

Dean, J.F., Middelburg, J.J., Röckmann, T., Aerts, R., Blauw, L.G., Egger, M., Jetten, M.S., de Jong, A.E., Meisel, O.H., Rasigraf, O. and Slomp, C.P.: Methane feedbacks to the global climate system in a warmer world. Reviews of Geophysics, 56, 1, 207-250, 2018.

Verbitsky, M.: Incomplete similarity of the ice-climate system, Earth Syst. Dynam. Discuss. [preprint], https://doi.org/10.5194/esd-2021-56, in review, 2021.

Verbitsky, M. Y. and Crucifix, M.: π-theorem generalization of the ice-age theory, Earth Syst. Dynam., 11, 281–289, https://doi.org/10.5194/esd-11-281-2020, 2020.

Verbitsky, M. Y. and Crucifix, M.: ESD Ideas: The Peclet number is a cornerstone of the orbital and millennial Pleistocene variability, Earth Syst. Dynam., 12, 63–67, https://doi.org/10.5194/esd-12-63-2021, 2021.

Verbitsky, M. Y., Crucifix, M., and Volobuev, D. M.: A theory of Pleistocene glacial rhythmicity, Earth Syst. Dynam., 9, 1025–1043, https://doi.org/10.5194/esd-9-1025-2018, 2018.

---

## Author Comment (AC9)

Dear Referee 2,

Thank you very much for the encouraging and insightful review. The following is our response to your comments.

**Comment:** This paper presents an interesting idea to derive a scaling law for global warming based on the physical fundamental physical properties of the climate system and the Buckingham-pi theorem for dimensional analysis. The scaling laws at two extreme conditions of positive or negative feedbacks dominating (complete similarity) are first derived, and then the case of incomplete similarity is discussed. I find this work novel and interesting, and a very good fit for an ESD Idea paper.

**Answer:** We appreciate your evaluation.

**Comment:** I have a number of questions and suggestions, mostly about the dimensional analysis part, which I hope the authors address.

- Line 57 and eq 3 (similarly, line 73 and eq 6): it is not clear to me how you can go from \phi(t/\tau_n, \lembda) to (t/\tau_n)^m in eq. (3). In Buckingham-Pi, the function \phi of some variables can be written as the product of each variable to an unknown power, but it is unclear to me why in eq. 3 there is no \lambda^q (q being another unknown power) in eq. 3 multiplied by the rest of the terms. This is my main comment/question about the method. Please clarify.

**Answer:** Your observation is correct. Indeed, $\lambda$^q may appear in equations (3) and (6). But since $\lambda$ and $\lambda$^q are constants, $\lambda$^q has been absorbed by the experimental constant $k$.

**Action:** We will clarify this reasoning in the revised version of the paper.

**Comment:** - Line 46: the dimension of \epsilon. I guess it is correctly written as sec^{-\lambda-1), however, it does not appear clearly as far as I see. Please take a look and clarify if possible

**Answer:** The dimension of $\varepsilon$ is [$^o$C sec^($-\lambda$-1)]. It looks like the pdf file of the preprint introduced some distortion in line 46 that may cause confusion.

**Action:** We will make sure that it is clear in the final version of the paper.

**Comment:** - Also you may want to clarify around line 50 that here there are 6 variables in (1), there are two fundamental dimensions (time and temperature), so Buckingham-pi states that there would be 4 dimensionless (pi) groups, as presented on lines 52-53.

**Answer:** Agreed

**Action:** This clarification will be included in the final version of the paper.

**Comment:** - line 22: the use of Buckingham-pi theorem in climate science has been rather limited, and the cited paper by Golitsyn is certainly a great example. You may want to also mention some of the recent papers using this approach to study the climate system, e .g.
+Chavas, D.R. and Emanuel, K., 2014. Equilibrium tropical cyclone size in an idealized state of axisymmetric radiative–convective equilibrium. JAS
+Yang, D. and Ingersoll, A.P., 2014. A theory of the MJO horizontal scale. GRL
+Nabizadeh, E., et al. 2019. Size of the atmospheric blocking events: Scaling law and response to climate change. GRL

**Answer:** Thank you for your suggestion. We agree that it will be helpful to our readers.

**Action:** Per editor's approval (number of references in ESD Ideas papers is limited to 12) we will be glad to include all your recommended references.

**Minor comments:**

- line 53: you may want to change this to ".... global temperature response T/(\epsilon t^{\lambda+1}) is a function of ....."
- I find the word "dimensionless" just sounding better than "adimensional" but it is up to you which one to use.
- line 99: m should be in the math mode
- line 16: is the word "similarity" in ".... in feedbacks similarity parameters ...." needed?

**Action:** All minor comments will be taken care of.

Mikhail Verbitsky and Michael E. Mann